# Learning Backchanneling Behaviors for a Social Robot via Data Augmentation from Human-Human Conversations

Michael Murray[1], Nick Walker[1], Amal Nanavati[1], Patrícia Alves-Oliveira[1], Nikita Filippov[1], Allison Sauppé[2], Bilge Mutlu[2], and Maya Cakmak[1]

[1]Paul G. Allen School of Computer Science and Engineering, University of Washington
[2]Department of Computer Sciences, University of Wisconsin–Madison
{mmurr, nswalker, amaln, patri, nikitaf, mcakmak}@cs.washington.edu
{asauppe, bilge}@cs.wisc.edu

**Abstract:** Backchanneling behaviors on a robot, such as nodding, can make talking to a robot feel more natural and engaging by giving a sense that the robot is actively listening. For backchanneling to be effective, it is important that the timing of such cues is appropriate given the humans' conversational behaviors. Recent progress has shown that these behaviors can be learned from datasets of human-human conversations. However, recent data-driven methods tend to overfit to the human speakers that are seen in training data and fail to generalize well to previously unseen speakers. In this paper, we explore the use of data augmentation for effective nodding behavior in a robot. We show that, by augmenting the input speech and visual features, we can produce data-driven models that are more robust to unseen features without collecting additional data. We analyze the efficacy of data-driven backchanneling in a realistic human-robot conversational setting with a user study, showing that users perceived the data-driven model to be better at listening as compared to rule-based and random baselines.

**Keywords:** social robots, backchanneling, data augmentation

## 1    Introduction

Social robots that can communicate with people naturally and effectively serve a wide variety of important daily life contexts, from companion robots for older adults [1], robots that help children with social-emotional development [2], or for robots that support teens to cope with stress and mental health challenges [3]. Verbal communication is the heart of many such applications of social robots. If social robots are to become a part of daily life, it is critical that talking to a robot feels natural and engaging. One way robots can remain active in a conversation while the human is speaking is through *backchanneling*. Backchannels, such as smiles, nods, or short utterances like "hmm", are cues that humans provide to each other during a conversation to indicate to the speaker that they are still actively listening and understanding [4]. Backchannels make a conversation feel smooth and natural [5]. When performed by a robot or virtual agent, backchannel responses have been shown to give users an increased perception of attentive listening, an increased feeling of rapport, and a desire to continue interacting [6, 7].

Research into the use of backchannels in human communication has found speaker cues that can elicit backchannel responses, such as periods of low pitch [8], rapidly changing intonations [9], and certain patterns of speaker gaze [10]. Many early approaches to programming backchannel behavior have been based on rules involving these cues [8, 11, 12]. Others have taken data-driven approaches, learning backchanneling behavior from datasets of human-human conversations. Over time, several data-driven approaches have been explored including decision trees [13, 14], sequential probabilistic models [15, 16, 17], and more recently, deep neural networks [18, 19, 20, 21]. Deep neural networks have shown promising performance in predicting when to backchannel, however, these approaches

5th Conference on Robot Learning (CoRL 2021), London, UK.

typically suffer from the generalization problem in which the learned backchanneling policies tend to overfit to human speakers that are seen in training data and fail to generalize well to previously unseen speakers. One of the major contributing factors to this problem is data scarcity. Learning backchanneling behavior requires annotated video datasets of human-human conversations which are expensive and time-consuming to collect. As a result, the datasets used to train backchanneling algorithms typically have a small number of human speakers. For example, the Interactive Emotional Dyadic Motion Capture Database (IEMOCAP) [22], which is the most commonly used dataset for learning backchanneling behaviors, consists of conversations from just 10 unique speakers. Due to the challenges of collecting sufficiently large datasets, it is a crucial problem to efficiently learn a more generalized policy for backchanneling without collecting additional data.

*Data augmentation* is a technique that has contributed to significant improvements in performance under conditions of data scarcity in domains such as image classification [23, 24, 25, 26] and speech recognition [27, 28, 29, 30]. With data augmentation, we produce new datapoints by augmenting existing datapoints, for example by warping the original input. This makes the training dataset effectively larger without collecting additional data, as multiple augmented versions of a single example can be used to train a model.

In this paper, we present a *data augmentation* scheme to learn more robust backchanneling behaviors for a social robot from human-human conversation data. We introduce a method that warps and masks portions of each human speaker's input features. Our experiments show that the technique results in improved performance on the metrics of accuracy and F1-score. Finally, we conducted a user study to examine the efficacy of a data-driven model for online human-robot interaction. As recent works have evaluated their models using pre-collected datasets, we contribute—to our knowledge–the first user study to examine a neural network based approach to backchanneling. Our results showed that, compared to rule-based and random baselines, the neural network approach elicits more natural and preferable interactions with a listening robot.

## 2 Related Work

**Modeling Human Backchannel Behavior**    The description, modeling, and understanding of human backchanneling behavior has been a subject of research since early comprehensive studies of non-verbal behavior in the 1970s [31, 4]. Early models sought to describe relationships between backchannel behaviors and speaker cues spanning gesticulation, gaze, syntactic markers and paralanguage [5, 32]. The ability to automatically detect speaker cues, like periods of low pitch or pauses, led to the first applications of rule-based backchanneling systems [8, 33, 11, 12]. The use of data driven methods such as decision trees [13, 14], sequential probabilistic models [15, 16, 17], and neural networks [18, 34, 19, 20, 21], has reduced the dependence on hand-designed rules.

**Embodied Backchanneling Agents**    The impact of backchanneling behaviors on interactions with virtual animated agents has been extensively studied [35, 36, 37]. Despite promising applications in educational or eldercare robotics, there are relatively few systems that have been designed to work with a physical robot. Lala et al. [38] evaluated an integrated backchanneling dialogue agent on ERICA, a realistic humanoid robot, finding that their approach was perceived as more natural and empathetic than rule-based alternatives. Inoue et al. [39] compared user responses to the system with those generated by a human operator and saw that their system matched the human control in basic listening skills but fell short in ratings for complex attributes like empathy. Park et al. [7] showed that children perceived a toy robot listener using a rules-based backchannel generation approach as more attentive compared to a random baseline.

**Data Augmentation**    Data augmentation has been applied to a wide variety of problems to reduce overfitting in neural networks. Image augmentations, such as horizontal flips and random cropping or translations, are commonly used in image classification and detection models [23, 24, 25, 26]. Krizhevsky et al. [24] notably achieved state-of-the-art results on image classification and showed that their augmentations reduced the error rate of their model by over 1%. Additionally, speech augmentations like speed perturbation, pitch adjustments, and additional noise are commonly used for automatic speech recognition [27, 28, 29, 30]. Recently, Park et al. [29] introduced SpecAugment, which achieves state-of-the-art performance on several speech recognition tasks by masking

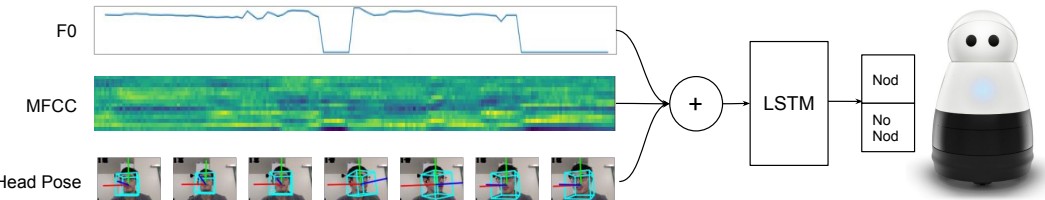

Figure 1: (From top to bottom) Sequences of F0, MFCC, and head pose features are extracted from the raw audio and video inputs. These features are concatenated and fed into an LSTM network which outputs a nod/no-nod decision for the robot.

the mel-spectogram representation of speech. Although data augmentation shows promise in many domains, it has not yet been applied to robot backchanneling, which we explore in this work.

# 3    Method

The goal of this work is to improve listening behaviors in a robot during a conversation with a human. Following the recent work of Ruede et al. [19], we model our policy as a recurrent neural network that maps sequences of input from the speaker to output actions on the robot. In this paper, we focus on *nodding* as a backchannelling modality because of it's prevalence and utility in human-human conversations [40, 41, 42], but we expect that the techniques we present will generalize to other backchanneling modalities. We aim to enable accurate prediction while maintaining the feasibility of deploying the model on computationally constrained robotics platforms.

**Acoustic Features**    To summarize the speaker's acoustic features we extract the fundamental frequency (F0) and 14th order mel-frequency cepstrum coefficients (MFCCs). F0 is an estimation of the speaker's pitch and the MFCCs represent the short-term power spectrum of the sound. Both features are commonly used in speech processing and can be calculated efficiently without introducing significant latency. We extract the features over 20ms windows and concatenate them to produce a single acoustic feature representation. This configuration significantly compresses the audio signal while maintaining the ability to distinguish prosodic cues and phonemes.

**Visual Features**    As a summary visual feature we extract the speaker's head pose, as an euler angle, using a fast facial landmark detector [43] and estimating the correspondence of points to a 3D head model by solving the perspective-n-point problem. This was informed by work showing gaze as a cue for backchannels [10]. To moderate noise inherent in this pipeline, we smooth the angle using a Kalman filter.

**Model**    We model the backchanneling policy as a long short term memory (LSTM) recurrent neural network [44], which is commonly used for modeling sequential data. At each step, the network receives the last two seconds of features as input and produces a binary output (Figure 1).

**Data Augmentation**    As we mentioned above, data scarcity is an issue when training neural networks for backchanneling tasks. Therefore, given a training dataset of the aforementioned input features and a binary output, we propose using data augmentation to add one new example for every example in the dataset.

Following Park et al. [29], we augment the MFCC features by warping them across time, masking blocks of utterances over time, and masking blocks of consecutive frequency channels. An example of these augmentations can be seen in Figure 2. We augment the F0 features by warping them across time and masking blocks of utterances over time. Finally, we augment the visual summary head pose feature by randomly warping the head pose in space and time, and by masking random blocks of head poses over time. These augmentations are intended to improve the model's robustness to partial loss of information (frequency, segments of speech, or head pose information) and deformations across time.

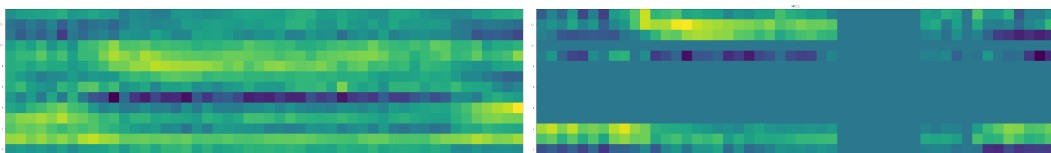

Figure 2: Spectogram visualization of an MFCC feature (left). And the same feature with warping and masking augmentation applied (right).

## 4  Data Collection and Systematic Evaluation

To evaluate our data-driven backchanneling models, we gathered a dataset of human non-verbal backchanneling behaviors. While we considered existing datasets such as IEMOCAP [22] and CCDb [45], no existing dataset met our criteria of being both unscripted and structured, with distinct speaker and listener roles. We believe these criteria will promote natural head nods that are more appropriate for a listening robot. This lead us to collect our own dataset to meet these criteria.

Using a custom peer-to-peer video chat tool, we recorded audio and video of conversational interactions between two people. Each interaction was structured around a prompt, with one person tasked to respond to the prompt while the other person was listening and performing natural backchannels. Every session was composed of one researcher and one participant selected from that researcher's social network. After obtaining consent, participants were given instructions on how to position themselves for optimal data collection—in a well-lit room, with no bright lights, and with their head in the middle of the frame. The researcher and participant then engaged in 10 interactions where they took turns prompting each other and listening to the other talk for 1-2 minutes. The prompts consisted of two random prompts from each of the following categories:

- Happy (e.g. "Tell me about your most fun memory from last summer")
- Neutral (e.g. "Tell me about your day yesterday")
- Teaching (e.g. "Teach me how to cook your favorite dish")
- Frustrated (e.g. "Tell me about a pet peeve of yours")
- Practice Talk (e.g. "You are giving a speech at your best friend, Sam's, wedding tomorrow. Practice that speech in front of me").

These categories were chosen to reflect the general scenarios in which we envision users speaking to an actively listening social robot. Accounting for interactions missing due to recording issues, we collected a total of 84 interactions from 12 participants (9 male, 3 female), 3 of whom were researchers. Interactions lasted 97.2 seconds on average (SD=31.2).

### 4.1  Annotation

We annotated listener nod behavior using a semi-automatic eye-position based scheme. The left eye keypoint position was extracted across the duration of the video using [43]. The range of this raw pixel position was then normalized to be between 0 and 1, and smoothed. The resulting signal was then processed to detect peaks and valleys. Portions of the signal with 3 or more peaks or valleys with no more than 50ms between them were marked as candidate nod windows. These candidates were then checked and adjusted manually using ANVIL [46]. After annotation was complete, an average of 21.5% of per-interaction listener time was labeled as nodding (SD=10.8).

### 4.2  Training

To train the RNNs we first split the dataset up into examples and labels where each example is a sequence of features extracted over the past 2 seconds and each label indicates whether the human subject nodded at that point in time. Because our dataset contains many more negative labels than positive labels, we account for the imbalance by assigning each class a weight that is inversely proportional to its frequency: $\alpha = \frac{n_+ + n_-}{2}$, $w_- = \frac{1}{n_-} \cdot \alpha$ and $w_+ = \frac{1}{n_+} \cdot \alpha$ (where $n_-$ is the number of negative examples and $n_+$ is the number of positive examples). We then split the dataset by individual speakers, and use 6-fold cross-validation to train the model. In each fold, the examples

Table 1: Comparison of LSTM, LSTM with data augmentation, and a rule-based baseline. Reported numbers are averages across all validation folds.

| Method | Training (Seen Speakers) | | | | Validation (Unseen Speakers) | | | |
| --- | --- | --- | --- | --- | --- | --- | --- | --- |
| | Acc. | Prec. | Rec. | $F_1$ | Acc. | Prec. | Rec. | $F_1$ |
| Rule-based (Ward) [11] | - | - | - | - | 0.60 | 0.20 | 0.14 | 0.23 |
| LSTM [20] | 0.80 | **0.31** | **0.29** | **0.60** | 0.62 | 0.18 | 0.16 | 0.24 |
| LSTM + Augmentation | 0.80 | 0.29 | 0.25 | 0.57 | **0.64** | **0.23** | **0.24** | **0.29** |

from two speakers are held out for validation and the remaining examples are used for training. We train the model by optimizing cross-entropy loss and perform early-stopping according to the loss.

### 4.3 Results

We compare the LSTM-based approach, with and without data augmentation, with a rule-based baseline. We chose Ward [11] as our baseline because it was the highest performing rule-based approach on our dataset. Table 1 shows the performance of the three models in terms of accuracy, precision, recall, and $F_1$-score. Precision is the probability that predicted head nods match the actual annotated human behavior, while recall is the probability that an annotated human head nod is predicted by our model. The $F_1$-score, which is the harmonic mean of precision and recall, is typically used as the key evaluation metric for imbalanaced datasets like ours [47].

We observe that both LSTM models outperform the rule-based model and the LSTM model with data augmentation achieves the highest accuracy, precision, recall, and $F_1$-score among all of the approaches. Figure 3 shows a plot of the training and validation $F_1$-scores across training epochs to further illustrate this reduction in the performance gap between training and validation. Additionally, in Figure 4 we present examples of qualitative improvements from two unseen speakers.

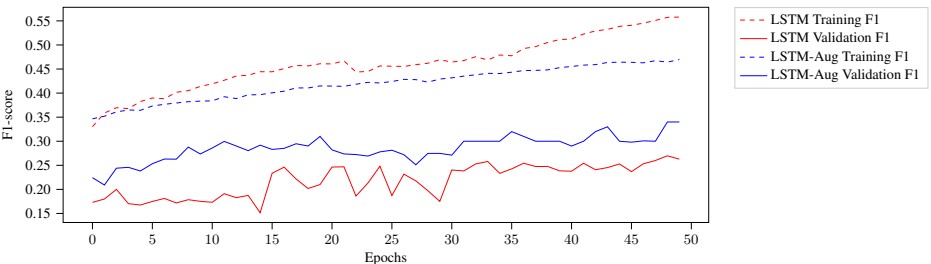

Figure 3: F1-score over 50 epochs for a selected training fold. Data augmentation achieves higher performance on the validation set due to less overfitting to the training set.

## 5 User Evaluation

While the systematic evaluation demonstrated the capacity for data-driven models to accurately predict backchanneling behaviors in human-human interactions, it does not capture the qualitative

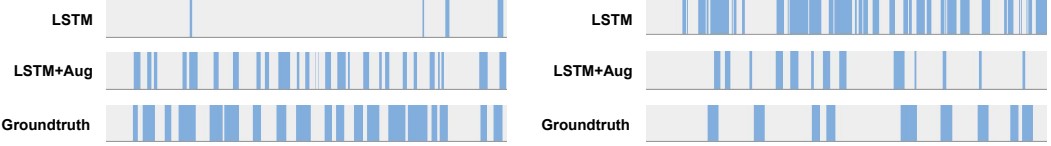

Figure 4: Visual timelines indicating nods over the duration of two example sessions that were both unseen during training. On the left, we can see that the LSTM nods too infrequently and the more robust data augmentation method is closer to the ground truth. On the right, the LSTM nods too frequently and again the data augmentation method looks closer to the ground truth.

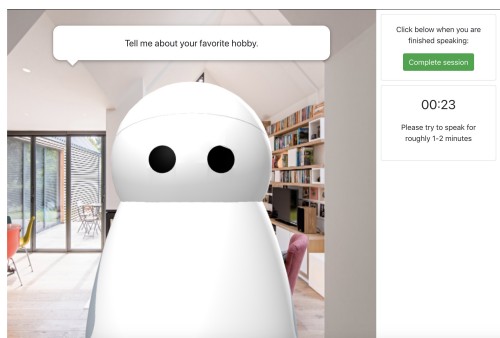
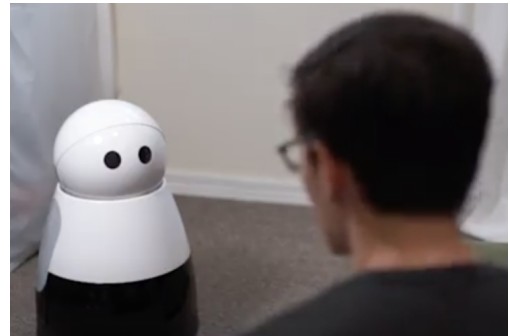

Figure 5: As part of the user study, participants talked to a virtual robot through an online interface, responding to a specific prompt (top). The robot nods when the model specifies.

Figure 6: While the user study employed a virtual robot to adapt to the COVID-19 pandemic, we also demonstrate that our method can be deployed on a physical Mayfield Kuri robot.

experience of interacting with a robot that uses the models to produce nodding behaviors. We investigated the effectiveness of these models in generating backchanneling behaviors on a robot through a user study in which participants talked to a virtual robot. We used a virtual robot to adapt to the COVID-19 pandemic.

## 5.1 Study Design and Procedure

Our study involved participants talking to a virtual Mayfield Kuri robot that uses either a learned model, a rule-based model, or a random baseline to determine when to nod. Specifically, the three conditions in our study were:

- **Learned** uses the best performing learned model, the LSTM model with data augmentation (Sec. 3), trained on the whole dataset. The model takes in audio and visual features, and the robot produces a nod whenever the model assigns a probability above 50% to the nod class.

- **Ward** [11] uses the aforementioned rule-based baseline. The model uses a rolling window to approximate the distribution of the speaker's pitch and nods upon detection of: (1) a region of pitch less than the 26th-percentile pitch level (2) continuing for at least 110 milliseconds (3) coming after at least 700 milliseconds of speech (4) no backchannels in the preceding 800 milliseconds (5) after 700 millisecond wait.

- **Random** produces a nod with nod duration sampled from a normal distribution fit to the durations in our collected dataset. Wait times between nods are sampled from an exponential distribution fit to the wait times in our collected dataset.

To allow for direct comparisons of the three conditions, we performed a within-participants study in which participants interacted with all three conditions. Each condition was presented as a different robot, named Robot 1-3. The order of conditions was counter-balanced. Mirroring our data collection procedure, this evaluation took place through the same custom online video communication interface. Participants were shown a simulated robot in a living space, intended to mimic a webcam display of a physical robot (see Figure 5). The robot exhibited naturalistic blinking and fidgeting idle behavior, and nodded when the model specified.

After reading instructions, participants went through a familiarization phase where they talked to a robot that had the idle behavior but did not nod. The interaction started with a question prompt from the robot. Participants were instructed to talk for 1-2 minutes responding to the prompt. The time elapsed was displayed on the screen for guidance and participants could move on at any point by pressing the "Complete session" button. After familiarization, they were taken to the study, where they interacted with each of the three models. The prompts for the study were taken from the *Frustrated* category of the data collection (Sec. 4) and the order of the prompts was kept constant. After the interaction with each model, participants were taken to a questionnaire that asked them about their perceptions of the robot and its listening and nodding behavior (Table 2).

Table 2: Survey questions from the evaluation questionnaire. The Likert statements required participants to choose their level of agreement with options ranging from 1-5 (strongly disagree to strongly agree). Frequency questions had a continuous slider between 0% of the time (never) and 100% of the time (always). Open ended questions were free text entry. Participants responded to this questionnaire after each condition.

| Likert Statements | Frequency Questions |
|---|---|
| • The robot was listening carefully.
• The robot's behavior was natural.
• The robot cared about what I was saying.
• The robot did not care about what I was saying.
• The robot was not listening to me.
• The robot understood what I was saying.
• I felt a close connection with the robot.
• Talking with the robot was similar to talking with a friend.
• Talking with the robot was similar to talking with a stranger.
• Talking with the robot was comforting. | • How often did the robot nod its head at an inappropriate time?
• How often did the robot miss head nod opportunities? |
| | **Open Ended Questions** |
| | • What was it like to speak with the robot?
• Describe positive and negative aspects of the robot's behavior
• Describe any similarities or differences between talking to the robot and talking to a friend |

After interacting with all three models and completing all three questionnaires, participants were asked to select the robot that seemed the most natural, that they enjoyed speaking to the most, that nodded the most, and that nodded the least. Finally, they were asked demographic questions. Participants for this study were recruited on Amazon Mechanical Turk, limited to users who had a greater than 95% approval rate and who were in the U.S., and were compensated $3 for their participation in this approximately 25 minute user study.

#### 5.1.1 Results

Our study was completed by 102 participants (53 male, 46 female, 3 other; ages 18-75). To group the Likert items we performed an exploratory factor analysis with promax rotation, yielding two factors shown in the appendix Table 7.1. We interpret the first factor as measuring the perceived nodding and listening skill of the robot. We interpret the second factor as measuring to what extent talking with the robot elicited feelings of comfort or closeness. The factors show good reliability with $\alpha = 0.88$ and $\alpha = 0.92$ respectively. We group the continuous frequency questions into a single scale measuring the perceived frequency of mistakes. Figure 7 summarizes participant responses on these scales. Figure 8 shows the distribution of participant preferences in forced-choice questions.

We used repeated measures ANOVA to measure the statistical significance of responses on the Likert and frequency scales. Using a significance threshold of $\alpha < 0.05$ for all statistical tests, there was a significant effect on the perceived nodding skill [$F(2, 204) = 9.86$, $p < 0.001$] and the perceived frequency of mistakes [$F(2, 204) = 4.56$, $p < 0.01$]. Post hoc comparisons using the Tukey HSD test indicated that the mean score of the learned model is significantly higher than the random model

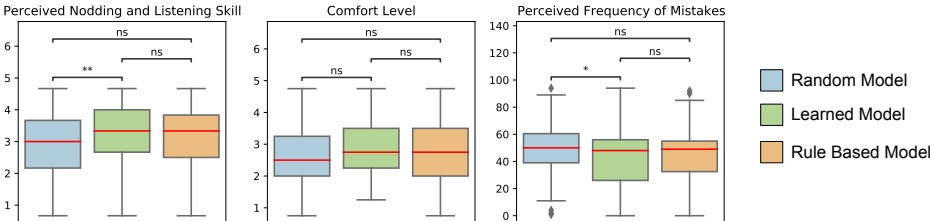

Figure 7: Box plots of participant responses to Likert scale and frequency questions in the user study (see Table 2 for exact questions). Whiskers represent 1.5 times inter-quartile range and the red line indicates the median value.

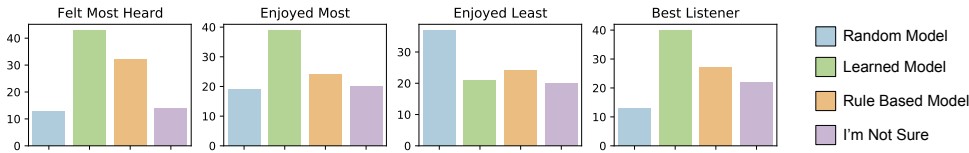

Figure 8: Histograms of participant responses to forced-choice questions asked after interacting with all conditions.

on both scales. The learned model also outperforms the rule-based model, although that difference is not significant.

User responses to the forced choice questions indicate a strong preference for the learned and rule-based models over the random model. The learned model was the most common choice for "Which robot made me felt most heard", "Which robot did you enjoy speaking to the most?", and "Which robot was the best listener?". Additionally, it was the least common choice for "Which robot did you enjoy speaking to the least?". While there was no significant difference between the learned and rule-based models on our Likert and frequency scales, there is a clear preference for the learned model over the rule-based model in the forced choice responses. This could indicate that the differences are too small to appear on the scales but still large enough to result in a clear preference or it could indicate that it was difficult for participants to express on a scale the differences they noticed when interacting with the virtual robot.

Some participants noticed the timing of nods, e.g., one said the learned model appeared "*better programmed to nod at more appropriate intervals*"; another said the rule-based model would "*nod at more appropriate times and didn't miss as many opportunities to nod*". While some participants were specific about the reasoning for their choice, as in the above quotes, others could not articulate behavioral reasons but rather answered based on how they felt after the interaction; e.g., comments about the learned model included that it "*seemed natural and normal*", "*felt like he was really listening*", and "*seemed more life like.*". Some users stated that they could not discern a difference; e.g. one user stated that there was "*no detectable difference*" and another stated "*it seemed like this robot had better timing when it came to nodding than before, but I'm not sure if that is actually the case or not*". Some users expressed optimism about the utility of a backchanneling robot; e.g. one user said "*I would use this to practice job interviews*" and another suggested to "*make these robots some kind of therapy app*". Other participants felt that the experience of talking to the robot was negative, highlighting important difficulties of social robot design; e.g. one person stated that "*Speaking to the robot felt dehumanizing.*".

## 6  Discussion

In this paper, we apply data augmentation to the task of learning non-verbal backchanneling behaviors from human-human interactions. Our experiments show that our data augmentation technique reduces overfitting and improves backchanneling prediction performance for unseen speakers. Data augmentation is not the only technique that has been developed to reduce overfitting, and an exciting direction for future work is to explore additional techniques such as pre-training and multi-task learning. Combining these techniques with reinforcement learning as in Hussain et al. [21] is another exciting future direction.

The results of our user study indicate that user perception of data-driven backchanneling behaviors is positive, with the data-driven approach being preferred in forced choice questions and eliciting nodding behavior that appears significantly more natural than a random baseline. Due to COVID-19 limitations our user study was conducted online with a virtual robot so we aim to repeat our evaluation with a physical robot in future work. Although our evaluation involved a virtual robot, the learned model was developed to work the same way on physical Kuri robots using on-board sensors and head nods. Future work could extend our approach to provide richer backchanneling with additional outputs like robot facial expressions and vocalizations. Because our work is focused on augmenting the input features, we expect that the techniques we have presented will generalize well to other backchanneling modalities. Further investigation of backchanneling within the context of larger human-robot interaction systems is another important direction for future work.

**Acknowledgments**

This work was supported by the National Science Foundation, collaborative awards IIS-1924435 and IIS-1925043 "Program Verification and Synthesis for Collaborative Robots."

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

# 7 Appendix

## 7.1 Likert item factor analysis loading matrix

| Variable | F1 | F2 |
|---|---|---|
| The robot was listening carefully. | 0.059 | **0.881** |
| The robot was not listening to me. | 0.241 | **0.939** |
| The robot's behavior was natural. | 0.404 | **0.471** |
| The robot understood what I was saying. | 0.403 | **0.489** |
| The robot cared about what I was saying. | **0.643** | 0.328 |
| The robot did not care about what I was saying. | **-0.558** | 0.305 |
| I felt a close connection with the robot. | **0.713** | 0.238 |
| Talking with the robot was similar to talking with a friend. | **0.955** | -0.026 |
| Talking with the robot was similiar to talking with a stranger. | **-0.839** | 0.251 |
| Talking with the robot was comforting. | **0.595** | 0.300 |

