# OpenReview forum: "Learning Backchanneling Behaviors for a Social Robot via Data Augmentation from Human-Human Conversations"
_robot-learning.org/CoRL/2021/Conference — CoRL2021 Poster_

### Official Review · Reviewer_uPNX · 2021-07-23

**Originality:** Fair
**Technical Quality:** Good
**Clarity Of Presentation:** Excellent
**Impact:** 3

**Recommendation:**

Weak Accept: I recommend accepting the paper, but will not argue for my recommendation if the majority of other reviewers have a different opinion.

**Summary:**

This paper is concerned with ways of generating backchanneling behaviors for robots (social cues like nodding appropriately placed in the midst of a conversation with a human). Data-driven methods tend to overfit to the (scarce) training set and cannot adapt appropriately to new speakers, so the paper presents a data augmentation method for the input speech and visual features that improves backchanneling behaviors without collecting additional data. The data augmentation schema warps and masks portions of each human speaker’s input features.

The work diligently collected their own dataset of unscripted but structured 2-person conversations, and used both audio and visual features for training a model that predicts when the agent should or shouldn’t nod. The paper evaluates the data augmentation strategy on this dataset with positive results, and later conducts a user study with a virtual robot that nods whenever the trained model specified it should do so.

**Issues:**

To improve the submission, I would suggest adding another couple backchanneling behaviors, reframing the paper to only focus on nodding, or making a strong argument for why the experiments presented are expected to generalize to other backchanneling behaviors. I also think rerunning the study to compare LSTMs with and without data augmentation would more directly study whether the proposed data augmentation is effective.

**Reviewer Expertise:**

Good: General knowledge of the area

**Strengths And Weaknesses:**

I thought the data collection and experimental analysis were really well done, with thorough evaluation and discussion. Additionally, the writing was clear and enjoyable, and the work was well placed within existing literature.

Overall, I liked the paper but there are a few sticking points that concern me/could improve this work. First, the paper discusses backchanneling behaviors as a whole but in reality only nodding is explored. I think the results would be stronger if they showcased a range of backchanneling behaviors (smiling or “hmm”s as well); otherwise, the work essentially boils down to a nodding predictor, which is not novel enough, in my opinion.

Secondly, while Section 4 compares LSTM with and without data augmentation, I was surprised to see that the same comparison didn’t exist in the user study. Since the main contribution of the paper seems to be using data augmentation to improve backchanneling, it seems natural that the study should focus on the effect of data augmentation, and not the effect of different types of predictors.

Lastly, unless I am interpreting Fig. 6 wrong, the study results don’t show a significant difference between the learned and the rule based model. I would have liked to see more discussion for why this is the case, especially since the forced choice questions in Fig. 7 indicate a significant difference.


**Summary Of Recommendation:**

Although (mostly) diligently done, the experiments in the paper are limited and I am reluctant to accept the paper without experiments on more backchanneling behaviors. Moreover, I am not convinced that the data augmentation method presented has a significant impact on people’s perception of the robot in conversation.

UPDATE AFTER-REBUTTAL: I thank the authors for their response. I have updated my score to a weak accept.

---

> ### Author Response · Authors · 2021-08-26
> **Thank you for reviewing our paper**
>
> Thank you for your feedback. We really appreciate your attention to detail and time spent on our submission.
>
> We appreciate your kind words about our data collection, experimental analysis, evaluation, and discussion. We are very glad that you found the writing clear and enjoyable.
>
> We also greatly appreciate your suggestions for improvement and in our revision we have addressed your feedback in the following ways:
>
> - We started with nodding because it is a well established problem studied in prior works (for example [15] and [20]) and because of its utility and prevalence in human-human conversations. We believe the techniques we present should generalize well to other backchanneling behaviors because we focus on augmenting the input features which should be beneficial to other downstream classifiers. In our revision we have added our rationale for focusing on nodding (Section 3), we have included more citations of prior research highlighting the importance and utility of nodding specifically (Section 3), and we have added our argument that the techniques should generalize well to other behaviors (Section 6).
> - We considered including multiple versions of the LSTM model in the user study, but we were concerned that having too many conditions would overload the participants and degrade the quality of their survey responses. Because none of the prior works had conducted user studies for any learned LSTM models, we needed to include non-learned models to compare to the state-of-the-art. Therefore, we compared the subjective experience of our best learned LSTM model to the subjective experience of the state-of-the-art non-learned baselines.  We then also added a model that draws randomly from distributions fit to the dataset, to compare to a trivial baseline.
> - You are correct that the responses to the Likert items only show a significant difference between learned and random. This could indicate that the differences are too small to appear on the scales but still large enough to result in a clear preference or it could indicate that it was difficult for participants to express on a scale the differences they noticed when interacting with the virtual robot. We have updated our revision to include this additional discussion in Section 5.1.1

---

### Official Review · Reviewer_kruZ · 2021-07-25

**Originality:** Good
**Technical Quality:** Good
**Clarity Of Presentation:** Good
**Impact:** 3

**Recommendation:**

Weak Accept: I recommend accepting the paper, but will not argue for my recommendation if the majority of other reviewers have a different opinion.

**Summary:**

- This work attempts to learn a policy of when a robot should nod, given the prosody and head orientation of a speaker
- The problem is of relevance because backchanneling is a crucial mechanism for maintaining the flow of conversation, and robots that are deployed as social agents will need such a capability
- The authors collect a dataset of conversations, and given annotations of when listeners nod, they train an LSTM on speech and head orientation data
- In order to combat the low generalizability of prior models to new speakers and provide sufficient data to train the model, the authors use data augmentation techniques such as random shifts and masks to prevent overfitting
- The authors's results show that the LSTM trained with data augmentation outperforms other baselines


**Issues:**

Barring the last point in the scope for improvement, I would like to see a better Likert analysis and some more clarification on the LSTM training.

**Reviewer Expertise:**

Good: General knowledge of the area

**Strengths And Weaknesses:**

**Strengths**

- The authors's motivation, formulation of the problem, and approach are straightforward and easy to follow
- The results in the paper are promising

**Scope for Improvement**

The biggest scope for improvement in this work lies in the presentation of the results:

- The Likert data could be better presented (show significance results in the plots!) and better analyzed (one should aggregate multiple items into a combined "scale"). For an excellent primer on how to approach Likert metrics, please see Schrum et al., HRI'20
- What was the loss function that was used for training the models? Additionally, was the loss used for early stopping (recommended) or accuracy (not recommended)? Clarification on this matter would be great
- (this is speculation) The F1 and accuracy scores are not that high (despite the promising results). Given the qualitative evaluation presented by the authors, might it be worthwhile to report an Intersection over Union-like metric instead (as is commonly used in computer vision)? If the authors wish to be truly rigorous, they can provide a correlation of the F1 metric with the IoU metric


**Summary Of Recommendation:**

Although the paper has weaknesses in its presentation of the results, they can be addressed by following some of the suggestions presented above. I am therefore, weakly in favour of accepting the paper into the technical program.

Post-Rebuttal Update:
I thank the authors for updating their paper and I find that it is more ready than before for publication. I am in favour of including this in the technical program.

---

> ### Author Response · Authors · 2021-08-26
> **Thank you for reviewing our paper**
>
> Thank you for your feedback. We really appreciate your attention to detail and time spent on our submission.
>
> We are glad that you found our motivation, formulation, and approach easy to follow; and we appreciate that you found the results in the paper to be promising.
>
> We also greatly appreciate your suggestions for improvement and in our revision we have addressed your feedback in the following ways:
>
> - We have updated the plots in Figure 6 to show significance results and the Likert items are now aggregated into two scales based on an exploratory factor analysis. We thank the reviewer for the helpful pointer to Schrum et al., HRI'20
> - Cross-entropy loss was used to train the models and we monitored the loss to perform early stopping. This information is now included in our revision (Section 4.2)
> - Although the F1 scores and accuracy for this task are lower than one might expect, they are in the same range as past works (for example [15] and [19]). The reason backchanneling tasks have lower F1 scores and accuracies is because of how we define ground truth. In our corpus (and most past works) we define the ground truth as times when the human listener nodded. But just because the human listener didn’t backchannel doesn’t necessarily mean that it was a bad time to backchannel. In Ward et al. the authors analyze the performance of their predictive rule and conclude that 44% of the incorrect predictions were cases where a backchannel could naturally have appeared. Although IoU is another valid measure of performance, it will also exhibit this problem because it’s also based on how we define our ground truth. Therefore, we continue to rely on F1-score because it is an accepted metric in prior works.

---

### Official Review · Reviewer_Uy2q · 2021-07-28

**Originality:** Very Good
**Technical Quality:** Very Good
**Clarity Of Presentation:** Excellent
**Impact:** 3

**Recommendation:**

Weak Accept: I recommend accepting the paper, but will not argue for my recommendation if the majority of other reviewers have a different opinion.

**Summary:**

This paper presents an approach for learning *backchannelling behaviors* for companion robots, an important aspect for cueing and signaling in multi-party interaction in general. Concretely, this paper presents a learned approach for deciding when to *nod* based on visual and speech/acoustic context from a human in conversation.

Critical to the proposed approach is intelligent application of data augmentation; obtaining backchanneling (nodding paired with videos of partner’s head pose, speech) data at scale is hard, and as such, naively training models results in robots with behaviors that are either overfit to speakers from the training set (and thus fail to generalize) or total collapse. By selectively masking and warping the input features (visual head poses, audio features) not only can we enrich the dataset, but we can learn better, more generalizable backchanneling behaviors.

The proposed model is simple, but effective (quite elegant!). Trained on data collected from 12 speakers primed with a diverse array of prompts, the resulting “virtual robot” experiment shows that not only is data augmentation critical to strong backchanneling performance (both qualitatively and quantitatively), but also out performs a strong, rule-based baseline.

A possible flaw in this work is the lack of deployment on a real robot, instead doing all experimentation with a virtual robot in a web interface; due to the COVID-19 pandemic, I think this is fair, and makes a lot of sense. I appreciate that the paper presents a clear path for deploying and evaluating with actual physical robots, and I am convinced the results will hold.


**Issues:**

[90 - 92] “Although data augmentation… it has not yet been applied to human-robot interaction” isn’t quite correct; several papers on human-robot interaction (e.g. learning from corrections, language instructions, etc.) all use data augmentation. Would temper this claim.

**Reviewer Expertise:**

Good: General knowledge of the area

**Strengths And Weaknesses:**

In terms of strengths, the problem is well-motivated, the proposed solution is elegant, the “key insight” (data augmentation) is also motivated, and proven to be necessary, and in general, the proposed system works convincingly from both a qualitative and quantitative perspective.

As a weakness; it does seem that backchanneling is just a smaller component of a larger HRI system (e.g., for human-robot dialogue, or assistive robotics). I understand the need to evaluate this specific behavior alone (and from the prior work cited, it seems that this is an established problem), but I do wonder if things change in the context of a broader system (e.g., is nodding really that important when a robot is equipped with the ability to provide dense feedback in terms of actual physical motion, or speech).

**Summary Of Recommendation:**

The proposed method is well-motivated, well-constructed, and the evaluation is strong. It’s clear that data augmentation is important for getting generalizable behavior from small backchanneling datasets, and that the proposed approach is a good step in solving this task.

---

> ### Author Response · Authors · 2021-08-26
> **Thank you for reviewing our paper**
>
> Thank you for your feedback. We really appreciate your attention to detail and time spent on our submission.
>
> We appreciate your kind words about our proposed method and evaluation.
>
> We also greatly appreciate your suggestions for improvement and in our revision we have addressed your feedback in the following ways:
> - In our revision we have corrected the scope of our claim to just “robot backchanneling”.
> - We agree that evaluation of this behavior in the context of a broader HRI system is an important direction for future work. We have updated the discussion (Section 6) to mention this point.

---

> ### Comment · Reviewer_Uy2q · 2021-09-03
> **Post-Rebuttal Update**
>
> Thanks to the authors for the responses. I'm happy to keep my score at a Weak Accept.

---

### Meta-Review · Area_Chair_7j5j · 2021-08-15

**Recommendation:** Accept (Poster)
**Confidence:** 4

**Metareview:**

This paper introduces a technique for automatically learning social backchanneling, in this case specifically the timing of nodding behaviors, from observation of human-human interactions.

Strengths:

The paper introduces a compelling learning approach which uses data augmentation techniques, such as random shifts and masks, to generalize the learned behaviors to new scenarios and speakers even from relatively little data.

The work is clearly motivated and well presented.

The experimental analysis is thorough and convincing.

Weaknesses:

Several technical details relating to the presentation of the algorithm and figures would improve the technical aspects of the paper.  These include treatment of Likert data and clarification of loss function.

---

> ### Author Response · Authors · 2021-08-26
> **Thank you for preparing the meta-review of our paper**
>
> Thank you for your feedback. We appreciate your time in organizing the reviews and preparing this meta-review.
>
> We are glad that you found the learning approach compelling, the work clearly presented, and the experimental analysis thorough.
>
> We appreciate your constructive feedback on the weaknesses of the paper. In our revision we have included more details to improve the technical aspects of the paper:
> - We have updated the plots in Figure 6 with significance results and the Likert items are now aggregated into two scales based on an exploratory factor analysis.
> - We now clarify the loss function and include details about early stopping in Section 4.2
>
> Additionally, we have addressed each reviewer individually in the posts below.

---

### Decision · Program_Chairs · 2021-09-13

**Decision:**

Accept (Poster)

**Comment:**

This paper introduces a technique for automatically learning social backchanneling, in this case specifically the timing of nodding behaviors, from observation of human-human interactions.

Strengths:

The paper introduces a compelling learning approach which uses data augmentation techniques, such as random shifts and masks, to generalize the learned behaviors to new scenarios and speakers even from relatively little data.

The work is clearly motivated and well presented.

The experimental analysis is thorough and convincing.

Weaknesses:

Several technical details relating to the presentation of the algorithm and figures would improve the technical aspects of the paper.  These include treatment of Likert data and clarification of loss function.